# Use of a Calving Blind That Imitates a Natural Environment

**DOI:** 10.3390/ani14081171

**Published:** 2024-04-13

**Authors:** Hannah E. Olsen, Kurt D. Vogel, Kate C. Creutzinger

**Affiliations:** Department of Animal and Food Science, University of Wisconsin—River Falls, River Falls, WI 54022, USA; hannah.olsen@my.uwrf.edu (H.E.O.); kurt.vogel@uwrf.edu (K.D.V.)

**Keywords:** maternal behavior, parturition, enrichment

## Abstract

**Simple Summary:**

Dairy cows seek seclusion at calving, but calving facilities typically do not provide them resources to do so. This study aimed to create a secluded environment in a group calving pen that mimicked natural features by providing cows with partial or no visibility to the larger pen. Approximately half the cows in the study calved in a blind. We also observed that cows spent time in a blind at times other than during calving. These results show that providing a varied environment can provide cows with increased choice over their environment, which can potentially improve their welfare.

**Abstract:**

Many indoor-housed cows isolate at calving when given the opportunity, and calving behaviors vary by blind and pen design. The objectives of this study were to determine if cows preferred calving in a visibly separated (blind) or an open area of a group maternity pen, and if there was a preference for the degree of seclusion provided by the blind (50% vs. 100% coverage). Two calving blinds were provided in a group calving pen, and the amount of visibility through the blinds was created using firehoses secured from the top of a metal frame that lined the entire front of the blind (100%) or with every other hose rolled up (50%). Holstein cows and heifers (n = 79) were enrolled into a dynamic group calving pen 21 ± 3 d before calving. Calving location, the difference in blind use prior to calving compared to a baseline period, and social behaviors were recorded using video observation. There was no difference in the number of cows that calved in or outside of a blind (28 vs. 37 calvings, respectively). Cows were more likely to calve in a blind during the day than at night and as the number of cows in the pen increased. For cows who calved in a blind, there was no preference for calving in the 50% or 100% blind (10 vs. 18, respectively). Providing a varied environment for intensively managed cattle can improve their welfare by allowing cows the opportunity to perform natural behaviors and choice over their environment.

## 1. Introduction

Research has recently begun to explore dairy cow preferences in calving pens and how their environment can impact behavioral changes around calving. It has been observed that dairy cows will seek isolation from other cows at calving using increased space and covered areas when they are available [1,2,3,4]. Calving pens are often built for ease of management, maintenance, and efficiency, rather than creating spaces for cows to separate and isolate themselves from their pen mates [5]. The reason for this is likely due to the need to maximize pen space per cow and the reduce fixed costs. Installing calving blinds is an easy way to create places for cattle to isolate without the need for additional pen space.

Dairy cattle in both indoor individual and group maternity pens seek isolation from other cattle during parturition by using blinds installed in the pens and distancing themselves from other cows when they have enough space [2,4,6,7]. Previous studies have utilized multiple variations of a place to isolate at calving in individual and group calving pens (i.e., calving blind). Some iterations of calving blinds have included a plastic road barrier filled with water in the center of a group calving pen [7]; individual calving areas inside a group calving pen that function like cubicles in an office building, with three full, solid sides and a fourth partial side facing the group pen [2,8]; a four-sided shelter with a “door” for study cows to move in and out of freely [4], and cubicle shaped blinds for an outdoor group calving pen that utilized two, interlocking gates covered in plywood that could be folded down to facilitate easy cleaning of the calving area [3]. While these blinds have successfully provided dairy cattle with a place to isolate at parturition, they may not be practical for use on dairy farms due to their difficulty to clean. Additionally, blind designs from previous studies lack the features of a natural environment.

Plywood has been the primary material for constructing calving blinds in past studies. However, solid walls prohibit cows from seeing their surroundings, unlike natural vegetation, which is generally composed of tall grasses and trees [9]. We aimed to imitate a natural environment by hanging firehoses to mimic the natural features of a calving site which allows varied visibility to the surrounding areas.

The objectives of this study were to determine if cows preferred calving in a calving blind or an open area in group maternity pens, and if cows preferred complete or partial visibility to the calving pen through a calving blind.

## 2. Materials and Methods

This experiment took place at the University of Wisconsin-River Falls (UWRF) Mann Valley Farm (River Falls, WI, USA) between January and July 2023. Cows were cared for in accordance with the protocol approved by the UWRF Animal Care and Use Committee (Protocol Number: 20-21-48101).

### 2.1. Animals, Housing, and Feeding

A total of 79 Holstein dairy cows (primiparous = 30, multiparous = 49) were enrolled in this study. This paper will refer to primiparous cows as animals that gave birth for the first time on this study. Once weekly on a fixed day (Monday), cows were moved into the group calving pen 21 ± 3 d before their expected calving date. The calving pen was a bedded pack and bedding (sawdust) was added on an as needed based on bedding moisture as determined by UWRF Mann Valley Dairy Farm staff. All bedding was completely removed and replenished once during the trial period.

The cows were fed an ad libitum total mixed ration (TMR) formulated for close-up cows once daily between 0800 and 1000 h and feed was pushed up once per day between 1800 and 2000 h. Water was available ad libitum via two automatic waterers in the feed alley of the treatment pen.

Sample size was determined with reference to a previous study that investigated similar outcomes [4]. We expected a 24% difference in blind use between the two options (blind use and open area). Using this effect size, we estimated that 67 dairy cattle would be needed to detect differences with 80% power at an α-level of 0.05.

### 2.2. Experimental Design

The study pen was 15.2 m long and 9.5 m wide, totaling 131.6 m^2^ of lying area. On average, there were 10 ± 2 cows (mean ± standard deviation) in the pen at a time (range: 6 to 16). The average lying space was 12.8 ± 2.9 m^2^/cow (range: 8.2 to 21.9 m^2^/cow). Two metal frames (4.6 m W × 3.05 m D × 3.05 m H) were constructed from 6.4 cm-diameter steel pipe with 0.5 cm wall thickness at the back of the study pen (Figure 1). The back wall of each blind was solid, and the two sides were covered by two layers of shade cloth. Cows could only enter the blinds through the front, which was covered by used firehose, allowing cows to push through the hose to enter (Figure 2). Used firehose was chosen as the material to use for the blind structure as it is durable and created a venetian blind effect that provided various amounts of visibility to the group pen from inside the blind. Firehoses were fastened across the front of the blinds such that each hose edge contacted the adjacent hose to create 100% blind density. To create the 50% blind density, every other hose was rolled up and secured to the top of the metal frame. The visual density of the blinds was switched after every 12 calving events. Cows in the study always had access to both the 50% and 100% dense blinds. The trial was divided into three periods to account for differences in weather: (1) 19 January 2023–18 March 2023, (2) 19 March 2023–17 May 2023, and (3) 18 May 2023–17 July 2023.

### 2.3. Inclusion Criteria

Cows were eligible for inclusion if they received no assistance at calving and delivered a single calf. Calving difficulty was scored on a three-point scale (0 = no assistance, 1 = calf pulled by hand or with chains, 2 = calf pulled with calf jack). Nine cows with a calving score > 0 were excluded from the study (primiparous = 5, multiparous = 4). Five cows that delivered twins were excluded from analysis (multiparous = 5).

### 2.4. Behavioral Observations

Behavior was continuously monitored using 6 digital video cameras (Swann HD Video Recorder, SWDVK-45804WL, Swann, Santa Fe Springs, CA, USA) and stored on hard drives (Western Digital WD Purple Surveillance Internal Hard Drive, Western Digital, San Jose, CA, USA) throughout the trial. One camera was mounted directly above each blind. Four cameras were mounted above and behind the feed alley pointed to look at the bedded pack treatment pen and feed alley. All areas of the pen were visible at all times. Individual cows were distinguished using their own markings that were identified on camera. All video observations were reviewed on the Swann HD Video Recorder that was used to record the videos, and data were recorded in Microsoft Excel (Microsoft Excel, version 16, Microsoft Cooperation, Redmond, WA, USA).

For all cows who met the inclusion criteria (n = 65), video was used to determine the calving time (defined as the time that the calf’s hips were expelled from the dam), calving location (open area, 50% blind, 100% blind), and the location of non-focal cows (open area, 50% blind, 100% blind). Calf sex (male vs. female) and breed (Holstein vs. dairy-beef cross) were also recorded. From the cows who met the inclusion criteria, a subset (n = 30) was pseudo-randomly selected for detailed behavior analysis (primiparous = 14, multiparous = 16, calved in the blind = 14, did not use a blind = 16) based on the methodology in Creutzinger et al. (2021) [7]. Cows who spent ≥ 90 min with their calf in the calving pen were automatically selected for analysis (n = 17; multiparous = 9, primiparous = 8, calved inside a blind = 9, calved in the open area = 8). The other 13 cows were selected to balance parity and blind use with the 14 originally selected using the random number generator in Microsoft Excel. One primiparous cow initially selected for detailed behavior analysis was removed after it was observed that she was locked in a holding alley for 5 h during the 12 h prior to calving. A multiparous cow that calved in the blind was selected using a random number generator to replace the removed cow.

Cows selected for detailed behavioral analysis were observed to determine blind use and social interactions with non-focal cows prior to calving. Focal cows were observed using instantaneous scan sampling at 10 min intervals during the 12 h prior to calving and the same 12 h period a week prior to calving (baseline) [5,7]. At each scan sample, it was recorded if the focal cow was in the open area or inside a blind (50% or 100%) and the number of non-focal cows in each blind. The total number of observations of cows within each location was summarized by hour (totaling 6 observations per hour) for the 12 h baseline period and the 12 h before calving. To determine if there was a difference between blind use leading up to calving and the baseline, the number of times a focal cow was observed in a blind per hour during the 12 h baseline period was subtracted from the number of times a focal cow was observed in a blind per hour during the 12 h before calving.

Cows selected for detailed behavioral analysis were continuously observed for 5 h prior to calving [10] via previously recorded video by a single trained observer (intraobserver reliability: *R*^2^ = 0.99). An ethogram of the recorded behaviors is provided in Table 1. For each social interaction, it was recorded if it occurred inside or outside the blind and if the focal animal was the actor (initiator) or reactor (receiver). Agonistic interactions included head butt, chase, and displacement from the blind. During analysis, we found that cows performed and/or received few to none of these behaviors. For data analysis purposes, head butts, chases, and displacement were summed into a single variable as the number of bouts per cow (agonistic behaviors). State behaviors (allogrooming) were summed as the number of bouts and total duration (s) per cow. For bout durations, 5 s was used as the bout criterion, meaning events were considered 1 bout for every 5 s of allogrooming. The bout ended when the behavior ceased for <5 s [11].

During video observations, it was observed that non-parturient cows commonly spent time in the blinds. We decided to include this descriptive information in the paper even though it was not one of the original objectives. The observations collected during focal cow baseline blind use were used to document non-parturient cow blind use. To calculate the non-parturient cow blind use, the number of observations of at least one cow present in a blind (50% or 100%) was summed within each hour of the day and divided by the total number of observations within that hour over a 24 h period, resulting in the average number of times a non-parturient cow was observed in a blind per hour from 0 to 2400 h.

### 2.5. Statistical Analysis

Descriptive data were calculated using Microsoft Excel. All statistical analyses were performed using SAS software (SAS Enterprise Guide, version 7.1; SAS Institute Inc., Cary, NC, USA). Raw data were visually assessed for data distribution and outliers using the UNIVARIATE procedure. Univariable analysis was conducted between the dependent and explanatory variables of interest. Explanatory variables associated with the dependent variable (*p* < 0.20) were included in a multivariable model. Manual backwards removal was performed and variables with *p* < 0.20 were retained in the final model. Statistical significance was declared at *p* < 0.05. Interactions were tested between biologically relevant variables and removed from the model if *p* > 0.05. No significant interactions were detected in the models unless otherwise stated. Model selection was based on best fit by examining Akaike Information Criterion and Bayesian Information Criterion or Pearson Chi Square/degrees of freedom.

A chi-square analysis was performed to determine if there was a difference in the number of cows that calved in a blind or the open area (PROC FREQ). To determine the likelihood of calving in a blind, we constructed a generalized linear mixed model with a binary distribution and link logit function using the GLIMMIX procedure. Variables tested for inclusion were parity (primiparous vs. multiparous), calf breed (Holstein vs. dairy-beef cross), time of day (“day” 0600 to 1759 vs. “night” 1800 to 0559), study period (1–3), number of blinds in use by non-focal cows (neither blind, 50% blind only, 100% blind only, or both blinds), calf sex (male vs. female), and total number of cows in the pen. Time of day and number of cows in the pen were retained as independent variables in the final model. Estimates were exponentiated in the model to interpret these predictions in the original scale of the data.

For cows that calved in a blind, we investigated if there was a difference in the number of cows who gave birth in the 50% or 100% blind, and if blind selection was affected by the presence of other cows in the blinds, using a Fisher’s Exact Test (PROC FREQ). The variable comparison included blind calving location (50% and 100%) × the blinds occupied by other cows at calving (neither blind, 50% blind only, 100% blind only, or both blinds).

A mixed linear model was used to assess the difference in blind use 12 h prior to calving compared to baseline blind use using the MIXED procedure. Univariable analysis was conducted between the dependent (blind use) and explanatory variables of interest, including hour before calving, if the cow calved in a blind or not, and parity. Hour before calving and if the cow calved in the blind were retained in the final model as fixed effects. Individual cow was included as a random effect and was set as the subject. Comparisons between hour before calving and if a cow calved in a blind were made using the PDIFF option with Tukey’s adjustment.

To investigate the effects of our factors and their relationship with agonistic behaviors (number of bouts per cow), a mixed linear model was performed with the MIXED procedure. Variables tested for inclusion were parity, time of day, study period, and total number of cows in the pen. Blind use was retained in the model as it was the primary outcome of interest. No variables met the criteria for inclusion into multivariate analysis. Very few cows performed and/or received allogrooming (6/30); thus, it was not statistically analyzed.

## 3. Results

### 3.1. Blind Use

A total of 28 (43%) of the 65 cows included in the study calved in a blind. Of the cows who calved in a blind, 18 (64.3%) used the 100% blind and 10 (36.7%) calved in the 50% blind. Of the cows who calved in the blinds, 18 (64.3%) were multiparous and 11/18 (61.1%) of the multiparous cows that calved in the blind used the 100% blind, while the remaining 7/18 (38.9%) gave birth in the 50% blind. Finally, 10/28 (36.7%) of cows who calved in a blind were primiparous; 7/10 (70.0%) that were primiparous calved in the 100% blind and the remaining 3/10 (30%) calved in the 50% blind.

There was no difference in the number of cows that calved inside or outside of a blind (ꭕ^2^ = 1.24, *p* = 0.26). Cows were more likely to calve in a blind if they calved during the day than at night (OR: 3.04, 95% CI 1.10–10.50, *p* = 0.03, Figure 3). The likelihood of calving in a blind also increased as the number of cows in the calving pen increased (slope = 1.38, SE = 0.18, *p* = 0.01). There was no difference in the number cows that calved in the 50% vs. 100% blind based on which blind was occupied at the time of calving (*p* = 0.15; Figure 4).

The number of times a focal cow was observed in a blind in the 12 h during the baseline and day of calving scan sampling is provided in Appendix A. There was a significant interaction between if cows calved in the blind and the hour before calving when compared to their baseline blind use (*p* = 0.04; Figure 5). Cows that calved in a blind spent more time in the blind during the hour before calving compared to cows that did not calve in the blind (*p* < 0.0001), but at no other time points. Cows who calved in the blind spent more time in the blind during the hour before calving than during the 12 h, 10 h, 9 h, 8 h, 7 h, 6 h, 5 h, 4 h, and 3 h prior to calving (*p* < 0.05). At least one non-parturient cow was observed in one or more of the blinds during 62% of the baseline scan sampling observations. Average blind usage by non-parturient cows fluctuated by time of day (Figure 6).

### 3.2. Social Behaviors

Descriptive statistics of the social behaviors recorded for each focal cow have been included in Appendix A. There was no difference in the number of agonistic behaviors during the 5 h before calving between cows that calved in the open area and those in the blind (least square means ± standard error: 9.3 ± 1.5 vs. 7.1 ± 1.6 bouts per cow, respectively; *p* = 0.33). A total of 20% of focal cows (6/30 cows, 1/6 calved in a blind vs. 5/6 calved in an open area) performed or received allogrooming in the 5 h before calving.

## 4. Discussion

The objectives of this study were to determine if dairy cows preferred to calve inside a calving blind or in an open area, and if they had a preference for the amount of visual seclusion provided by a calving blind at calving. Cows in this study did not exhibit a preference for calving inside or outside the blinds. However, cows were more likely to calve inside a blind during the day and as the number of cows in the group calving pen increased. Cows who used the blind to calve had greater blind use before calving than during a baseline period, which indicates a calving-specific preference. Interestingly, non-parturient cows also spent time in the blinds, which may suggest that cows will choose to spend time in different environments at different points of their lactation.

In this study, there was no calving location preference between outside and inside a blind (37 vs. 28 calvings, respectively). The percent of cows in our study that calved in a blind was similar to those in previously published studies for cows housed in group maternity pens. For example, 37% of cows in a group maternity pen calved next to a blind in the center of a pen [7], 34% of pair-housed cows calved in a shelter [4], and 47% of cows calved in a three-sided hide in an outdoor group maternity pen [3]. Our blind was different in design when compared to these studies, as it aimed to represent a natural setting, such as trees and tall grasses, more closely by providing some visibility through the blind with the aim of providing cows with a more preferrable calving blind Providing novel designs is important to understanding cow preference because preference tests are limited by the choices provided by researchers, and none may be desirable for animals in the studies [12,13]. Additionally, dairy cattle exhibit individual variations in preference for resources such as outdoor access [10] and the voluntary use of a shower for cooling [14]. A somewhat consistent rate of cows who calved in a secluded and open environment in this and previous studies may suggest that cows have individual preferences for calving environments too. Offering multiple variations of seclusion of calving environments (e.g., none to full coverage) may allow cows to choose a calving location most suitable to them.

The likelihood of calving in a blind was affected by environmental factors, including the number of other cows in the pen and time of day. Cows were more likely to calve in a blind as the number of other cows in the pen increased. This is inconsistent with results from studies that found fewer cows calved in a blind as the stocking density of the pen and the cow to blind ratio increased [2,3]. The authors of these studies suggested that fewer cows may have calved in a blind when the number of cows to blinds was greater due to increased competition for blind use. We speculate that the results from this study show the inverse relationship of previous studies regarding the number of cows in the pen due to the size of the blind in our study, which allowed more than one cow to fit comfortably inside the blind. Having increased space inside the blind may have reduced competition for blind use at calving. Further, it is unclear how cows chose between different social environments at calving (i.e., one cow vs. many cows present). Additional preference tests and a greater understanding of preferred social partners for cattle may help to elucidate cows’ preferences for the social environment at calving.

Cows were also more likely to calve inside a blind if they calved during the day than at night. This is similar to the results found by Proudfoot et al. (2014b) [4], who found that cows preferred to calve in a blind but only if they calved during the day. Comparatively, an outdoor study found that cows were less likely to calve in a blind than an open area, but a majority of these cows moved into a blind with their calves, and were more likely do so if they calved in early evening and the middle of the night than in early morning [3]. It may be possible that increased human and other cow activity could have influenced the cows’ preference to calve in the seclusion of the blind during the day. Additional consideration to the placement of calving pens and activity around them may be warranted to best suit cows’ preferences at calving. When evaluating a location for a calving pen, farms may want to consider finding a low-traffic area in the barn, compared to a higher traffic area, to place the calving pens. If calving pens are placed in a high-traffic area, farms should consider providing cows with a blind and or a visual break to feel more secluded during the calving process.

Of the cows who calved in a blind, there was no difference in the number of cows who calved in the 50% and 100% visually dense blinds (18 and 28, respectively). Rørvang et al. (2017) [1] also found that cows in individual calving pens did not have a preference between three different amounts of blind seclusion in a group maternity pen that was visible from the individual calving pens (50% tall and narrow, 50% short and wide, and 75% tall and wide); only cows who had a longer calving duration calved in the most secluded pen area. Additionally, there was no relationship between blind use for calving and if the blinds were occupied by other cows. Based on descriptive data, we believe it is possible that this study was underpowered to detect differences between blind density preference, and if blind selection was affected by the presence of other cows in the blinds. This is a limitation of the study, and differences may have been detected if more cows were enrolled. Therefore, it is suggested that future studies use a larger sample size.

Cows who calved in the blind spent more time in the blind from baseline observations during the hour before calving compared to cows who did not calve in a blind. Compared to baseline blind use values, cows who calved in a blind also increased their blind use during the hour prior to calving compared to the previous hours leading up to calving. These results are similar to those of Proudfoot et al. (2014b) [5] and Creutzinger et al. (2021) [7], who found that cows increased their use of the provided blind space in the hours before calving. Greater blind use at calving indicates the cows who calved in a blind had a specific preference at calving, and did not exhibit a clear preference for the blind at all times compared to cows who calved in an open area.

Throughout the study, non-parturient cows also used the blinds. In total, there was at least one cow present in one of the blinds during 62% of observations. Over a 24 h period, the number of times at least one cow was observed in a blind varied between 40% and 83% of observations per hour. Blind use other than leading up to and during calving suggests a need for increased environmental variation at other points of the cows’ lactation. Both dairy and beef cattle have sought out environmental protection indoors using a blind and cover outdoors when ill [6,15]. While the periparturient cows in this study did not show signs of illness, the use of calving blinds by non-parturient cows suggests that cows will use a varied enrichment when provided. Providing environmental enrichment, such as a blind, may provide increased opportunities for agency that gives cows choice over their environment.

## 5. Conclusions

Cows housed in a group maternity pen preferred to use a secluded calving area as the number of cows in the maternity pen increased and if they calved during the day, which suggests blind use may be affected by other environmental factors. Blinds were occupied by both periparturient and non-parturient cows during the study, suggesting that environmental enrichment will be used by cows when provided.

## Figures and Tables

**Figure 1 animals-14-01171-f001:**
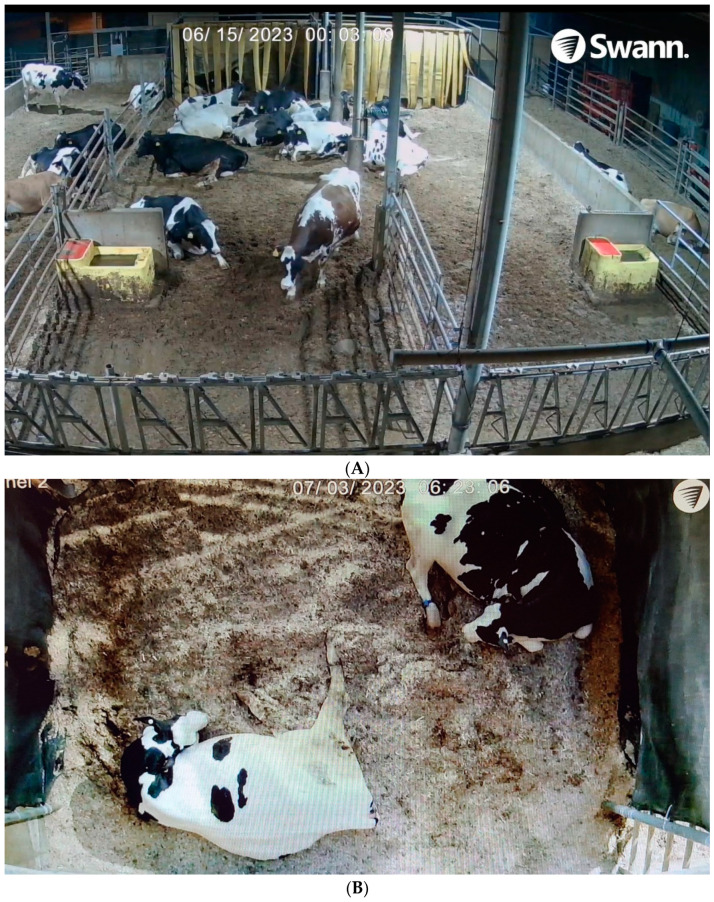
(**A**) Video images of the study pen area looking over the feed alley onto the open area of the study pen. Both blinds are visible at the back of the pen. At the time this image was taken, the 50% blind was on the left side of the study pen and the 100% blind was on the right side of the study pen. (**B**) Video image with a top-down view into a blind with two cows inside.

**Figure 2 animals-14-01171-f002:**
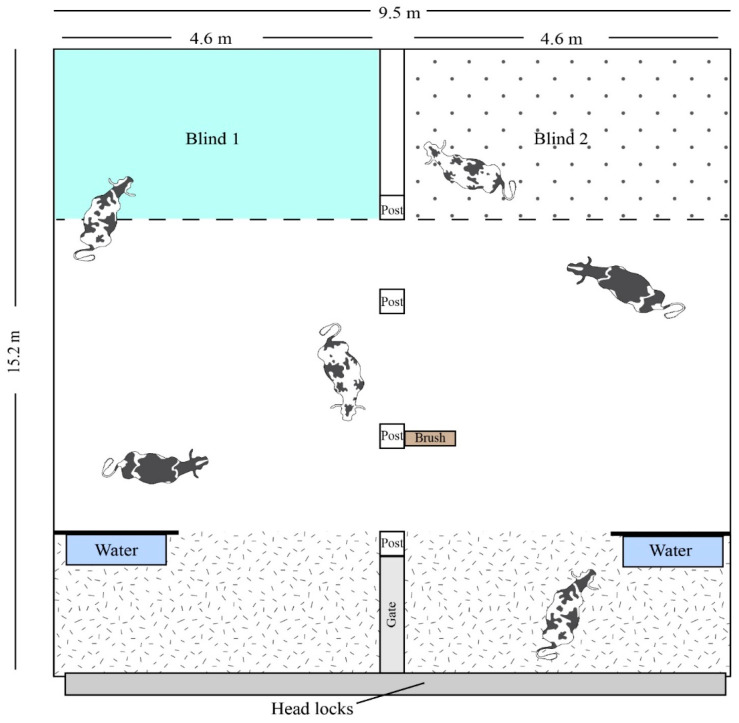
Top view of the experimental pen (15.2 m × 9.5 m lying space). Cows had access to both blinds during the entirety of the study period. Blind density was switched every 12 calvings. Dashed lines indicate the fire hose that comprised the front of the blind, the only way the blind could be accessed. The hashed area represents the feed alley. Feed was only provided in the head lock area. Cow image attribution: Jason C. Fisher, University of California Los Angeles (ian.umces.edu/media-library).

**Figure 3 animals-14-01171-f003:**
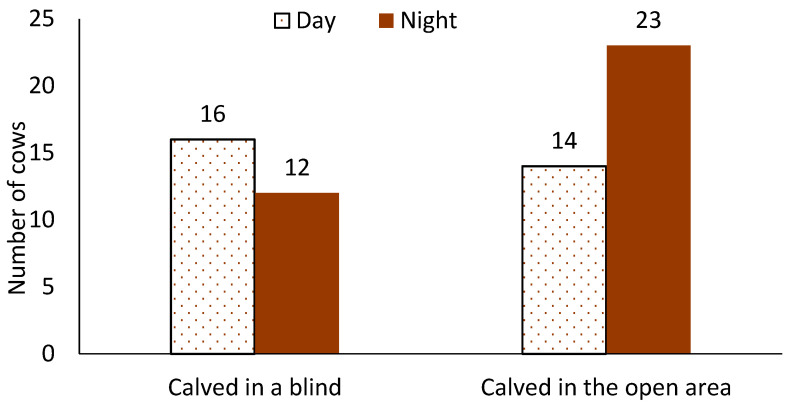
Number of cows who calved inside a blind (n = 28) or calved in the open area (n = 37) during the day (0600–1759) and at night (1800–0559).

**Figure 4 animals-14-01171-f004:**
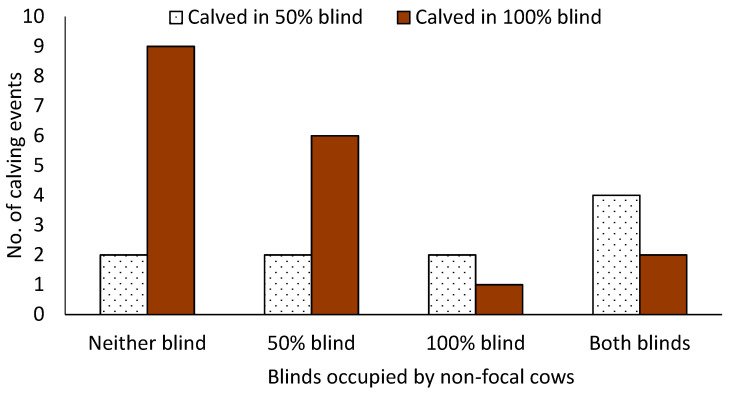
Relationship between blind occupancy by non-focal cows and blind use by focal cows during calving events (n = 28). The horizontal axis is labeled as neither blind (no cows present in either blind except the focal cow during calving), 50% blind (non-focal cow(s) present in the 50% blind during calving), 100% blind (non-focal cow(s) present in the 100% blind during calving), and both blinds (non-focal cow(s) present in both the 50% and 100% blinds during calving).

**Figure 5 animals-14-01171-f005:**
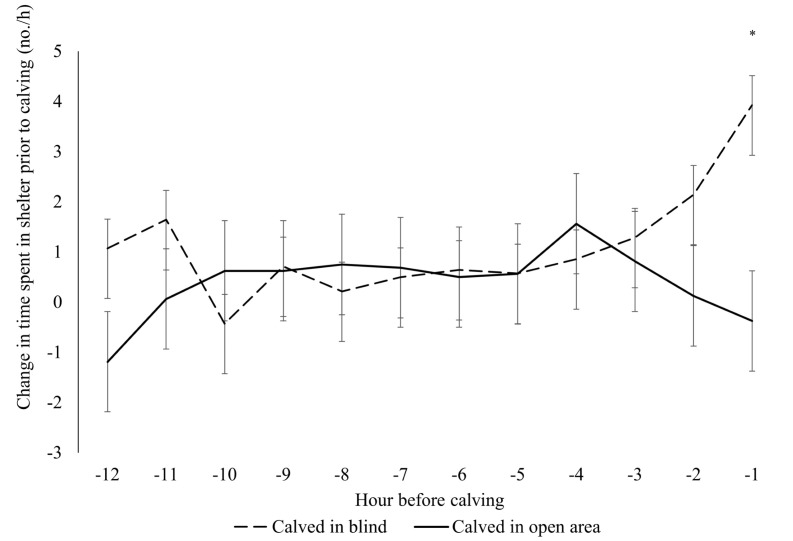
The difference in the number of observations of a cow in a blind per hour during the 12 h before calving compared to a 12 h baseline period recorded 1 wk before calving. Video was scan-sampled every 10 min for a subset of focal cows (n = 30) selected for detailed behavioral analysis that calved in either a blind (50% or 100%; n = 14) or in the open area (n = 16). An asterisk (*) denotes a difference (*p* < 0.05) between cows that calved in a blind and in the open area.

**Figure 6 animals-14-01171-f006:**
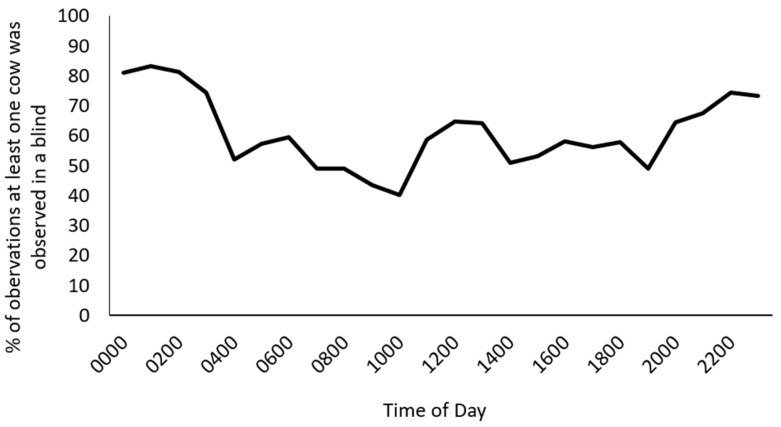
Descriptive data of the percentage of observations in which at least one non-parturient cow was recorded as present in one of the blinds per hour (50% or 100%).

**Table 1 animals-14-01171-t001:** Description of continuously recorded social behaviors of the focal cows during the 5 h continuous observation period before calving. All observations were recorded as the number of bouts and allogrooming was also recorded as duration in time (s).

Behavior	Description
Allogrooming	Cow was observed licking any part of another cow’s body
Head butt	Cow’s head makes contact with any part of another cow’s body
Chase	Cow moves quickly towards another cow, without contact, causing cow to walk or run away
Displacement from the blind	Cow makes physical contact with another cow that is standing or lying

## Data Availability

Data are contained within the article and Appendix A.

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
