# Peer review of "Use of a Calving Blind That Imitates a Natural Environment"

_animals, 2024, doi:10.3390/ani14081171_

Round 1
Reviewer 1 Report
Comments and Suggestions for Authors
please see attached comments

Author Response
Reviewer #1
Abstract:
Line 14-16: the description of objectives should be more complete in the abstract.
AU: Thank you for your comment. The objectives were edited to read. “The objectives of this study were to determine if cows preferred calving in a visibly separated area (blind) or in the open area of a group maternity pen, and if there was a preference for the level of seclusion provided by the blind (50% secluded or 100% secluded).” (lines 13-15)
- Introduction:
Line 54-56: should reference the authors associated with this statement.
AU: This statement was made by the authors. We have changed it to read, “they may not be practical”, because this has not been studied specifically (lines 47-48).
Line 63-64: I think the example of chickens will not be suitable for this article, review or
delete.
AU: This statement has been removed.
- Materials and Methods:
Line 84-86: complete with more information - in addition to the time at which the TMR
was administered to the cows, they should mention the amount of TMR per cow.
AU: It now states that the ration was available ad libitum (line 70)
- Results:
Line 301: Despite what they refer to regarding social behavior, it would be important to
present some graph or table with the frequency of behaviors identified in the materials
and methods (table 1).
AU: A table with descriptive data summarizing the social interactions for each cow included in the focal cow social behavior analysis has been included as Supplemental File 2. It is referred to in lines 260-261.

Reviewer 2 Report
Comments and Suggestions for Authors
I was excited to review this manuscript! It covers an interesting and important topic that deserves more attention. I appreciated your group’s novel approach to enhancing environmental complexity for calving. I am recommending that this manuscript be accepted with revisions.
Introduction
I’d suggest removing LN 64-66 and using this piece of information in the Discussion when you talk about the challenges with preference testing later around LN 326. Where it is at now undercuts the work that you are about to present versus if you talked about this point later on, you could use the information to strengthen why your study design included multiple options.
Methods
I love that the premise of your experiment is to mimic the natural environment that cows would seek when they are seeking cover during calving, and how they are motivated to still be able to see out of the covered area that they choose. I was hoping to learn more about the reason that you specifically chose to use a firehose. I saw in the Acknowledgements that they were donated but I think the reader would benefit from more context as to why this material. Is it easy for cattle producers to obtain or would this be a potential barrier for producers to implement at their own farms?
Also, I was personally wondering if the weight of the firehose or having to walk through them deterred any cows from using it. Did you see that there were cows who never entered that area or did all cows in the pen at some point use the space? I think adding this information into the Results would be nice for future researchers and producers to understand if they would want to use firehose material when seeking to create more environmental complexity into their calving pens, OR if there’s a different material that would be better suited for cows OR if that needs to be explored further.
I loved the inclusion of Figure 1 and Figure 2. What a nice schematic! It really helped me visualize what your group was doing.
LN 141: Can you include the reasoning for using a subset of the cows for more detailed behavior analysis? I’m assuming it is because of the labor that continuous sampling takes? However, then I was surprised that scan sampling was used. Why was scan sampling not done for all cattle? Also, how was the sample size of 30 determined for the detailed behavior analysis?
LN 162-166: I did not understand what was being explained here. Could you add more context for the reader?
Was scan sampling all conducted by the same individual or multiple individuals? If multiple individuals, who trained these observers and what was the inter-rater reliability? Please include this information in the Methods section.
What software (e.g., BORIS, Noldus, VMC Media Player) was used to view the videos? What software (e.g., BORIS, Noldus, Excel spreadsheet, etc.) was used to code behaviors? Please include this information in the Methods section.
LN 174 – 178: I think that you were trying to convey that only frequency was recorded for agonistic interactions (head butt, chase, displacement from blinds) and that frequency and duration were recorded for allogrooming. This might be a difference in how different ethologists talk about behavior coding. I was trained with reference to Martin & Bateson to view events as frequency only behaviors. In this case, head butt, chase, displacement from blind and to view states as behaviors that have both a duration and frequency. I would consider a bout of allogrooming to be a state behavior. If you agree, please adjust the language in these sentences to reflect this. Also, for agonistic interactions, I think it would be beneficial to the reader for you to say which behaviors you included or lumped together as agonistic interactions.
Results
I thought it was an intriguing and useful finding that cows who calved during the day were more likely to calve using a blind compared to cows who calved at night. However, I am having a difficult time understanding that from Figure 3 and wondering if there is a better way to convey this information visually. It almost seems like Figure 3 doesn’t support this finding and that there isn’t much of a difference between cows choosing to calve under the blind vs. in the open space during the daytime hours.
Figure 4 also has me scratching my head since there is no difference in the number of cows that calved in the 50% vs. 100% blind based on which blind was occupied at the time of calving. However, when I view Figure 4, I would think that there was a difference as it appears to me that cows calved more frequently in 100% blind when no other cows were present in the blinding areas. Again, I’m wondering if there is a better way to present your finding to the reader visually or if there’s a better way to explain what is being demonstrated in the Figure through the legend. It also just may not be necessary to include a figure to demonstrate this point.
LN 263 – I think there may be a word or a few words missing in this sentence.
Figure 5 – The reader could benefit from a more detailed explanation of what is being demonstrated in this figure so the figure can stand alone.
Discussion
LN 328-332: I was a bit surprised to learn that previous researchers have discovered that lower stocking density meant that fewer cows would use the blind. I would think that the opposite (AKA what you found) would be true as they’d want more privacy. Can you expand on this a bit more? I know you had a bit about the size of your blind, but then that sort of confused me more because you mentioned that it fit multiple cows comfortably and I was under the impression that the cows wouldn’t want to birth in the covered area with an “audience” so to speak.
LN 334 – 347: What a beautiful paragraph! I loved reading this portion.
LN 355 – 357: I had to go back to earlier in the paper at this point, because I knew you included a wonderful section about how you determined your sample size estimation. This line undercuts your work a bit, could you perhaps rephrase this to discuss that now looking back at how you conducted your sample size estimation and having three options for calving, that you would recommend future work uses a larger sample size or something of this effect? Give yourself some credit as you were doing something novel! You don’t know what you don’t yet know!
LN 362 – 365: Very strong section! Loved it.
LN 373 – 376: YES! Loved reading this portion, as well. Thank you for advocating for greater environmental complexity for our intensively kept animals.
Comments on the Quality of English Language
There were a few sentences where words appeared to be missing or that would benefit from a re-write for clarity to the reader. I noted these in my reviewer comments.
Author Response
Introduction
I’d suggest removing LN 64-66 and using this piece of information in the Discussion when you talk about the challenges with preference testing later around LN 326. Where it is at now undercuts the work that you are about to present versus if you talked about this point later on, you could use the information to strengthen why your study design included multiple options.
AU: To address this comment, we moved the discussion of preference tests and added a more robust discussion on inter-cow variation that may influence the difference in behavior (lines 287-293)
Methods
I love that the premise of your experiment is to mimic the natural environment that cows would seek when they are seeking cover during calving, and how they are motivated to still be able to see out of the covered area that they choose. I was hoping to learn more about the reason that you specifically chose to use a firehose. I saw in the Acknowledgements that they were donated but I think the reader would benefit from more context as to why this material. Is it easy for cattle producers to obtain or would this be a potential barrier for producers to implement at their own farms?
AU: Thank you so much for your kind comment! We included a description of how we selected fire hoses for this project. The main aspects we selected it for were durability and the ability to change visibility between the blind and the pen (lines 85-87)
Also, I was personally wondering if the weight of the firehose or having to walk through them deterred any cows from using it. Did you see that there were cows who never entered that area or did all cows in the pen at some point use the space? I think adding this information into the Results would be nice for future researchers and producers to understand if they would want to use firehose material when seeking to create more environmental complexity into their calving pens, OR if there’s a different material that would be better suited for cows OR if that needs to be explored further.
AU: This was an interesting and pertinent question since we talk about the challenges of preference tests. We were limited on time to review if all the cows in the study used the blind. To answer your question, we summarized the number of times a cow was observed in a blind during the 12 hour scan sampling period on the day of calving and the week prior to calving for the cows included in the focal cow analysis. This has been provided in Supplemental File 1. We think it will be a valuable addition to the manuscript. It is referenced in lines 241-242
I loved the inclusion of Figure 1 and Figure 2. What a nice schematic! It really helped me visualize what your group was doing.
AU: Thank you so much!
LN 141: Can you include the reasoning for using a subset of the cows for more detailed behavior analysis? I’m assuming it is because of the labor that continuous sampling takes? However, then I was surprised that scan sampling was used. Why was scan sampling not done for all cattle? Also, how was the sample size of 30 determined for the detailed behavior analysis?
AU: We chose to observe a subset of focal cows for detailed analysis due to the amount of time that would have been required for continuous observation. Both scan sampling and the 30 cow subset was based on Creutzinger et al., 2021ab and Proudfoot et al., 2014. These studies are cited in line 150.
LN 162-166: I did not understand what was being explained here. Could you add more context for the reader?
AU: Upon rereading this, we agree that it was confusing. We have removed this text and believe it makes more sense now.
Was scan sampling all conducted by the same individual or multiple individuals? If multiple individuals, who trained these observers and what was the inter-rater reliability? Please include this information in the Methods section.
AU: Good point. We have defined that a single trained observer watched all the video and provided the r-squared value for intraobserver reliability testing (line 158).
What software (e.g., BORIS, Noldus, VMC Media Player) was used to view the videos? What software (e.g., BORIS, Noldus, Excel spreadsheet, etc.) was used to code behaviors? Please include this information in the Methods section.
AU: The following statement was included “All video observations were reviewed on the Swann HD Video Recorder that was used to record the videos and data was recorded in Microsoft Excel (Microsoft Excel, Microsoft Cooperation, Redmond, WA, USA).” (lines 130-133)
LN 174 – 178: I think that you were trying to convey that only frequency was recorded for agonistic interactions (head butt, chase, displacement from blinds) and that frequency and duration were recorded for allogrooming. This might be a difference in how different ethologists talk about behavior coding. I was trained with reference to Martin & Bateson to view events as frequency only behaviors. In this case, head butt, chase, displacement from blind and to view states as behaviors that have both a duration and frequency. I would consider a bout of allogrooming to be a state behavior. If you agree, please adjust the language in these sentences to reflect this. Also, for agonistic interactions, I think it would be beneficial to the reader for you to say which behaviors you included or lumped together as agonistic interactions.
AU: Thank you for this comment. The following statement was added “Agonistic interactions included head butt, chase, and displacement from the blind. During analysis, we found cows performed and/or received little to none of these behaviors. For data analysis purposes, head butts, chases, and displacement were summed into a single variable as the number of bouts per cow (agonistic behaviors). State behaviors (allogrooming) were summed as the number of bouts and total duration (s) per cow.” (lines 161-165)
Results
I thought it was an intriguing and useful finding that cows who calved during the day were more likely to calve using a blind compared to cows who calved at night. However, I am having a difficult time understanding that from Figure 3 and wondering if there is a better way to convey this information visually. It almost seems like Figure 3 doesn’t support this finding and that there isn’t much of a difference between cows choosing to calve under the blind vs. in the open space during the daytime hours.
AU: We understand the reviewer’s point. In an attempt to clarify this outcome, the design of Figure 3 has been converted from a stacked column chart design to a clustered column chart design. Additionally, we reworded the conclusion to read, “cows were more likely to calve in a blind during the day than at night”. (lines 226-227). We believe this is factual because more cows calved in the blind during the day than at night (16 vs. 12, respectively) and more cows calved in the open area at night than in a blind (23 vs. 12, respectively). Hopefully, this addresses the reviewer’s question.
Figure 4 also has me scratching my head since there is no difference in the number of cows that calved in the 50% vs. 100% blind based on which blind was occupied at the time of calving. However, when I view Figure 4, I would think that there was a difference as it appears to me that cows calved more frequently in 100% blind when no other cows were present in the blinding areas. Again, I’m wondering if there is a better way to present your finding to the reader visually or if there’s a better way to explain what is being demonstrated in the Figure through the legend. It also just may not be necessary to include a figure to demonstrate this point.
AU: Again, we agree with the reviewer. It is our hypothesis that the study was underpowered to detect a difference in the number of cows that calved in each blind and if those blinds were occupied by another cow. We have addressed this as a limitation in the manuscript (lines 327-330)
LN 263 – I think there may be a word or a few words missing in this sentence.
AU: Good catch! The sentence now reads “Video was scan sampled every 10 minutes for a subset of focal cows (n = 30) selected for detailed behavioral analysis that calved in either a blind (50% or 100%; n = 14) or in the open area (n = 16) (lines 249-253).
Figure 5 – The reader could benefit from a more detailed explanation of what is being demonstrated in this figure so the figure can stand alone.
AU: We have revised the Figure 5 caption to read as follows, “The difference in the number of observations a cow was observed in a blind per hour during the 12 h before calving compared to a 12 h baseline period recorded 1 wk before calving. Video was scan sampled every 10 minutes for a subset of focal cows (n = 30) selected for detailed behavioral analysis that calved in either a blind (50% or 100%; n = 14) or in the open area (n = 16).” (lines 264-256). Hopefully, this addresses the reviewer’s comment.
Discussion
LN 328-332: I was a bit surprised to learn that previous researchers have discovered that lower stocking density meant that fewer cows would use the blind. I would think that the opposite (AKA what you found) would be true as they’d want more privacy. Can you expand on this a bit more? I know you had a bit about the size of your blind, but then that sort of confused me more because you mentioned that it fit multiple cows comfortably and I was under the impression that the cows wouldn’t want to birth in the covered area with an “audience” so to speak.
AU: This is an interesting comment. We have added additional context in hopes to clarify this for the reviewer and provide greater background to contextualize this outcome (lines 298-306)
LN 334 – 347: What a beautiful paragraph! I loved reading this portion.
AU: Thank you so much!
LN 355 – 357: I had to go back to earlier in the paper at this point, because I knew you included a wonderful section about how you determined your sample size estimation. This line undercuts your work a bit, could you perhaps rephrase this to discuss that now looking back at how you conducted your sample size estimation and having three options for calving, that you would recommend future work uses a larger sample size or something of this effect? Give yourself some credit as you were doing something novel! You don’t know what you don’t yet know!
AU: This is very generous of you. We have reworded this section to read as follows, “therefore, it is suggested that future studies use a larger sample size.” (line 330)
LN 362 – 365: Very strong section! Loved it.
AU: Thank you!
LN 373 – 376: YES! Loved reading this portion, as well. Thank you for advocating for greater environmental complexity for our intensively kept animals.
AU: We really appreciate this feedback. We too believe this is important.
Reviewer 3 Report
Comments and Suggestions for Authors
The study by Olsen et al. investigated the use of blinds to create secluded areas within a calving pen for dairy cows. The study was particularly intriguing due to its innovative approach of using discarded firehoses to create separation from the open area, closely resembling natural environments. The inclusion of varying levels of blinds also added an interesting dimension to the research. This area of study has garnered increasing interest in recent years, particularly concerning the welfare of dairy cows during transition period. The approach used in this study has practical applications that can be easily implemented, potentially encouraging farmers to use similar materials in calving pens. The manuscript was well-written, easy to follow, and quite engaging. The Materials and Methods section was particularly well-described. Below are some minor comments provided to further strengthen the manuscript.
Line 12: The abstract needs to be reduced to about 250 words to comply with the journal guidelines.
Line 68: Consider having a separate paragraph for the objective at the end of the introduction section.
Figure 5: The title of the y-axis overlaps with the axis values.
Author Response
Line 12: The abstract needs to be reduced to about 250 words to comply with the journal guidelines.
AU: The abstract has been revised and now has less than 250 words (lines 12-26)
Line 68: Consider having a separate paragraph for the objective at the end of the introduction section.
AU: We have moved the objectives to their own paragraph (lines 54-56)
Figure 5: The title of the y-axis overlaps with the axis values.
AU: Good catch! Figure 5 is adjusted so the title of the y-axis does not overlap with the axis values.